BIBR1532 inhibits proliferation and enhances apoptosis in multiple myeloma cells by reducing telomerase activity

Zhang Yuefeng 2016150206@jou.edu.cn 1
Yang Xinxin 1
Zhou Hangqun 2
Yao Guoli 1
Zhou Li 3
Qian Chunyan 4
1 Department of Hematology, First People’s Hospital of Linping District , Hangzhou , Zhejiang , China
2 Medical School, Hangzhou Normal University , Hangzhou , Zhejiang , China
3 Department of Oncology, First People’s Hospital of Linping District , Hangzhou , Zhejiang , China
4 Clinical Laboratory, First People’s Hospital of Linping District , Hangzhou , Zhejiang , China
Qin Jiangjiang
Electronic publication date: 2023 Nov 8
Publication date: 2023
Volume: 11
Electronic Location ID: e16404
Received 2023 Aug 2; Accepted 2023 Oct 13
Copyright: ©2023 Zhang et al.
Copyright year: 2023
Copyright holder: Zhang et al.
License: This is an open access article distributed under the terms of the Creative Commons Attribution License, which permits unrestricted use, distribution, reproduction and adaptation in any medium and for any purpose provided that it is properly attributed. For attribution, the original author(s), title, publication source (PeerJ) and either DOI or URL of the article must be cited.
License URL: https://creativecommons.org/licenses/by/4.0/

Keywords: BIBR1532, Multiple myeloma, Cell proliferation, Apoptosis, Synergistically

Funding: Health Science and Technology Research Project of Hangzhou 2017B27 Clinical Research Fund Project of Zhejiang Medical Association 2017ZYC-A43 This study was supported by the Health Science and Technology Research Project of Hangzhou (No: 2017B27) and the Clinical Research Fund Project of Zhejiang Medical Association (No: 2017ZYC-A43). The funders had no role in study design, data collection and analysis, decision to publish, or preparation of the manuscript.

==============================
Background

Multiple myeloma (MM) is a rare haematological disorder with few therapeutic options. BIBR1532, a telomerase inhibitor, is widely used in cancer treatment and has promising outcomes. In this study, we investigated the efficacy and mechanism of action of BIBR1532 in MM.

Methods

K562 and MEG-01 cells were cultured with BIBR1532 at different concentrations. After 24 and 48 h, cell survival was analyzed. Next, these cells were cultured with 25 and 50 µM BIBR1532 for 48 h, then, cell proliferation, apoptosis, and the expression of the telomerase activity related markers were tested by 5-Ethynyl-2′-deoxyuridine (EdU) staining, flow cytometric analysis, western blot and quantitative real-time PCR (qRT-PCR), respectively. Expression of Bcl-xL, Bad, Survivin, phosphorylation of PI3K, AKT, mTOR, ERK1/2, and MAPK were tested via western blotting. Further experiments were conducted to evaluate the synergistic effects of BIBR1532 and doxorubicin (Dox) or bortezomib (Bor).

Results

BIBR1532 inhibited K562 and MEG-01 cell survival in a dose- and time-dependent manner. In addition, BIBR1532 hindered cell proliferation while promoting apoptosis, and this effect was enhanced by increasing the BIBR1532 concentration. Moreover, BIBR1532 inhibited TERT and c-MYC expression, PI3K, AKT, mTOR phosphorylation, and facilitated ERK1/2 and MAPK phosphorylation. Additionally, BIBR1532 combined with Dox or Bor showed synergistic effects in MM treatment.

Conclusion

BIBR1532 inhibits proliferation and promotes apoptosis in MM cells by inhibiting telomerase activity. Additionally, BIBR1532 combined with Dox or Bor exhibited synergistic effects, indicating that BIBR1532 may be a novel medicine for the treatment of MM.

Introduction

Multiple myeloma (MM) is an uncommon hematological disorder that results from the clonal proliferation of plasma cells (Wang et al., 2021b). The neoplastic growth of these plasma cells results in several complications such as anemia, renal failure, and fractures (Ikoma et al., 2019). Over the last 30 years, advances in treatment options have significantly enhanced the life expectancy and prognosis of patients with MM (Akhtar et al., 2022). Nevertheless, the development of therapeutic resistance in most patients has resulted in a lack of cure, ultimately resulting in poor outcomes. Hence, there is a tremendous need to develop new strategies to achieve effective treatments for MM.

Several studies have demonstrated that cell cycle aberrations are involved in MM pathogenesis (Kerros et al., 2009). Unregulated cell proliferation and apoptosis enable tumor cells to invade and metastasize into normal tissues (Shen et al., 2019). Thus, suppressing cell proliferation and promoting apoptosis are important strategies for MM therapy (Thakur et al., 2018). Telomerase is a highly regulated enzyme, whose activity is critical for cell growth and death. Telomerase activation results in uncontrolled cell proliferation or even tumorigenesis. Moreover, approximately 90% of immortalized and malignant cells exhibited abnormal telomerase activity (Wang et al., 2021a), including MM cells (Kumar et al., 2018). Conversely, the loss of telomerase activity causes cells to enter a proliferative crisis and then die (Volleth et al., 2020). In addition, Li et al. (2020) reported that the inhibition of telomerase activity hindered the migration and invasion of esophageal squamous carcinoma cells. Given the essential role of telomerase activity in cell growth and death, the identification of drugs that can inhibit telomerase activity is a promising strategy for MM therapy.

The non-nucleoside compound BIBR1532 represses telomerase activity by specifically binding to the active site of hTERT and inducing senescence in human cancer cells (Giunco et al., 2020). BIBR1532 has a potent capacity to inhibit tumor cell growth in many cancers such as ovarian, chondrosarcoma, breast, lung, and germ cell tumors (Lavanya et al., 2018). In addition, research reported that BIBR1532 can interfere with cell proliferation, invasion, as well as survival in feline oral squamous cell carcinoma, by reducing telomerase activity and decreasing EGFR, ERK phosphorylation and matrix metalloproteinases (MMPs). Furthermore, it revealed that BIBR1532 may have potential as an adjuvant in combination with radiotherapy and other anti-cancer drugs in breast cancer, lung cancer, and acute promyelocytic leukemia (Altamura et al., 2020). However, the specific function and mechanism of action of BIBR1532 in MM remain unknown.

Hence, in this study, we performed cellular experiments to investigate whether BIBR1532 can hamper cell proliferation and promote apoptosis in MM, by inhibiting telomerase activity via the regulation of the PI3K/AKT/mTOR and ERK1/2 MAPK pathways. Moreover, given that single-drug treatment may cause drug resistance, we conducted experiments to explore whether BIBR1532 exerts a synergistic effect with doxorubicin (Dox) and bortezomib (Bor), thus providing a solid experimental basis for the utilization of BIBR1532 as an underlying therapeutic drug for MM.

Materials & Methods

Cell culture

Human myelogenous leukemia (K562, iCell-h118) and megakaryocytic leukemia (MEG-01, iCell-h307) cell lines were purchased from iCell Biosciences (China). All cells were incubated at 37 °C in Roswell Park Memorial Institute (RPMI) 1640 medium with 10% fetal bovine serum (FBS), penicillin (100 U/mL), as well as streptomycin (100 µg/mL) under 5% CO2.

Drug preparation

BIBR1532, Dox, and Bor were obtained from Selleck (S1186), Sigma (D9891), and Shanghai Source Leaf Biological Technology Co. Ltd. (B34605), respectively. BIBR1532 storage solution was solubilized in dimethylsulfoxide (DMSO) and kept at −20 °C for further use. Dox was dissolved in sterile water, and Bor was dissolved in phosphate-buffered saline (PBS) to prepare stock solutions.

Methyl thiazolyl tetrazolium (MTT) assay

The MTT assay was performed to assess the viability of MM cells. Briefly, the cells were cultured in 96-well plates (5000 cells/well) and stimulated by BIBR1532 (0.75–100 µM), Dox (100 nM) and Bor(10 nM) alone or in combination. After 24 or 48 h incubation, 5 µL MTT solution (ST316; Biyuntian Biotechnology Co., Ltd., Shanghai, China) was dropped into each well and the cells were cultured for 2 h at 37 °C. The cells in the control group were untreated. The reaction was stopped by dissolving the formazan crystals in DMSO. A microplate reader (CMaxPlµs) was applied to read the optical density value at 570 nm.

5-Ethynyl-2′-deoxyuridine (EdU) staining

Cell proliferation was evaluated by EdU staining. Briefly, cells plated on 96-well plates were stimulated by BIBR1532 (25 or 50 µM) for 48 h. Subsequently, cells were cultured with 2 µL EdU solution (C0078s; Biyuntian Biotechnology Co, Ltd., Shanghai, China) for 4 h, based on the manufacturer’s instructions. Following fixation in 95% ethanol, the cells were permeabilized with Triton X-100 (0.3%) and washed with PBS. Thereafter, the click reaction solution was added to the wells, and the cells were cultured for another 30 min in the dark. Following incubation with Hoechst 33342 (1 mL) to stain the nuclei, the cells were viewed and photographed under a microscope.

Apoptosis assay

Apoptosis was monitored by flow cytometry. First, all cells were treated with BIBR1532 (25, 50 µM), Dox (100 nM) and Bor (10 nM) alone or in combination. After 48 h, cells were collected and cultured with 5 µL of Annexin V-FITC, as well as 10 µL of PI (556547; BD Biosciences, San Jose, CA, USA) in darkness. Apoptotic cells were measured within 1 h using flow cytometry (NovoCyte; Agilent, Shanghai, China).

Western blot

Western blotting was performed to investigate the expression of cellular proteins. Briefly, after treatment with the corresponding compounds for 48 h, all cells were cleaved to obtain total protein, and protein concentration was assessed using a bicinchoninic acid (BCA) assay. The proteins were separated using 10% sodium dodecyl sulfate-polyacrylamide gel electrophoresis (SDS-PAGE) and transferred onto nitrocellulose membranes. After incubation with 5% skim milk, the membranes were probed at 4 °C overnight with the primary antibodies as follows: anti-Bcl-xL (AF6414), anti-Bad (AF6471), anti-Survivin (AF6017), anti-hTERT (DF7129), anti-c-MYC(BF8036),anti-p-PI3K (AF3241), anti-PI3K (AF6241), anti-p-AKT (AF0016), anti-AKT (AF6259), anti-p-mTOR (ab109268), anti-mTOR (AF6308), anti-p-ERK (4370T), anti-ERK (4695T), anti-p-MAPK (AF4001), anti-MAPK(AF6456), and anti-GAPDH (AF7021). The following day, secondary antibodies (1:6000; 7046 or 7074; CST, Danvers, MA, USA) were added to the membranes, and incubated for 1 h at room temperature. Finally, protein bands were detected using enhanced chemiluminescence (ECL) reagents (610020-9Q, Clinx Science Instruments, China) and quantified using ImageJ software. All primary antibodies used in this study were diluted 1:1000, except for GAPDH, which was diluted 1:10000. In addition, all primary antibodies were provided by Affinity, except p-mTOR from Abcam and p-ERK and ERK from CST.

Telomerase activity assay

Telomerase activity was determined using a quantitative real-time PCR (qRT-PCR) detection kit (KGA1029H; KeyGEN Biotech, Nanjing, China). Telomerase from cell extracts added telomeric repeats to the end of an oligonucleotide substrate, which was subsequently amplified by PCR. The PCR products were labeled with SYBR Green and quantified by measuring the increase in fluorescence intensity. Cell lysates were prepared and analyzed with the same amounts of protein for each telomeric repeat amplification protocol (TRAP) assay. All reactions were run at least three times with a negative control for every sample and heat inactivation was included before every assay. TSR, an oligonucleotide with the same sequence as telomere primers, was used to generate a standard curve. The primer sequences used in this study are listed in Table 1.

Statistical analysis

The data from this study was analyzed with SPSS 16.0, and is displayed as mean ± SD. Multiple group comparisons were performed using one-way analysis of variance (ANOVA) and Tukey’s tests. The Kruskal-Wallis H test was applied when the variances were not homogeneous. p < 0.05 was considered as statistically significant.

Results

BIBR1532 inhibited MM cell survival

To investigate the effect of BIBR1532 on MM cell survival, K562 and MEG-01 cells were stimulated with varying doses of BIBR1532 (0.75–100 µM). After incubation for 24 or 48 h, cell survival was determined and the IC50 value was calculated. The MTT results presented in Fig. 1 show that for K562 and MEG-01 cells, BIBR1532 at various concentrations suppressed cell survival, and the inhibition efficacy was significantly elevated with increasing doses (p < 0.05). Moreover, cell survival decreased after prolonged treatment. Based on the IC50 value obtained from the MTT assay, 25 and 50 µM of BIBR1532 and a 48 h treatment time were selected for the following study.

Table 1 qPCR primers.

Gene	Forward Primer	Reverse Primer	
Human TERT	GCCGATTGTGAACATGGACTACG	GCTCGTAGTTGAGCACGCTGAA	
Human GAPDH	CATCTTCTTTTGCGTCGCCA	TTAAAAGCAGCCCTGGTGACC	

Figure 1 BIBR1532 decreased cell survival in multiple myeloma cells.

BIBR1532 repressed MM cell proliferation

To evaluate the role of BIBR1532 in MM cell proliferation, K562 and MEG-01 cells were cultured with BIBR1532 and EdU staining was performed. As shown in Fig. 2, for both K562 and MEG-01 cells, relative to the controls, the proportion of EdU-positive cells was markedly reduced after treatment with BIBR1532 (p < 0.05). Moreover, the proportion decreased with increasing BIBR1532 concentrations.

Figure 2 BIBR1532 reduced the cell proliferation of multiple myeloma cells.

BIBR1532 promoted MM cell apoptosis

To evaluate the effect of BIBR1532 on MM cell death, flow cytometry was performed. It was observed that for both K562 and MEG-01 cells, the apoptosis rate was significantly elevated upon treatment with 50 µM BIBR1532 for 48 h; moreover, BIBR1532 also effectively facilitated the MEG-01 cells apoptosis at a concentration of 25 µM (Fig. 3A). As expected, western blot results revealed that compared with the controls, the expressions of Bcl-xL and Survivin in both K562 and MEG-01 cells were decreased. Meanwhile, the expression of Bad was increased after BIBR1532 stimulation, especially at a concentration of 50 µM (p < 0.05, Fig. 3B).

Figure 3 (A–B) Acute apoptosis was promoted by BIBR1532 in multiple myeloma cells.

BIBR1532 reduced TERT and c-MYC expressions for MM cells

To confirm the repressive effect of BIBR1532 on telomerase activity in MM cells, TERT and c-MYC expression was tested after MM cells were treated with the agent. It can be observed that for both K562 and MEG-01 cells, TERT gene expression was significantly decreased after stimulation with 50 µM of BIBR1532, while 25 µM of BIBR1532 significantly decreased TERT mRNA expression for MEG-01 cells (p < 0.01, Fig. 4A). In addition, BIBR1532 also reduced hTERT and c-MYC protein expressions in both K562 and MEG-01 cells, especially at a dosage of 50 µM (p < 0.01, Fig. 4B).

Figure 4 (A–B) BIBR1532 downregulated TERT and c-MYC expressions in multiple myeloma cells.

BIBR1532 suppressed PI3K/AKT/mTOR but facilitated ERK1/2 MAPK pathway in MM cells

To detect the effect of BIBR1532 on the PI3K/AKT/mTOR and ERK1/2 MAPK pathways, western blotting was performed to measure PI3K, AKT, mTOR, ERK1/2, and MAPK phosphorylation in MM cells. As displayed in Fig. 5A, for K562 cells, after treatment with BIBR1532, PI3K and AKT phosphorylation were robustly reduced while ERK1/2 phosphorylation was profoundly increased, regardless of whether the cells were treated with 25 or 50 µM of BIBR1532 (p < 0.01). Furthermore, 50 µM of BIBR1532 also effectively decreased mTOR phosphorylation and elevated MAPK phosphorylation in K562 cells (p < 0.05). Similarly, when MEG-01 cells were stimulated by 25 or 50 µM of BIBR1532, PI3K, AKT and mTOR phosphorylation were significantly decreased while ERK1/2 phosphorylation was markedly raised (p < 0.05). Moreover, 50 µM of BIBR1532 also significantly upregulated MAPK phosphorylation in MEG-01 cells (Fig. 5B).

Figure 5 (A–B) BIBR1532 repressed PI3K/AKT/mTOR while promoting ERK1/2 MAPK pathway for multiple myeloma cells.

Combination of Dox or Bor with BIBR1532 synergistically suppressed viability and promoted apoptosis of MM cells

To clarify whether BIBR1532 exhibited synergy in combination with Dox and Bor, we first measured the anti-proliferative and pro-apoptotic effects using MTT and apoptosis assays. As illustrated in Figs. 6A–6B, for both K562 and MEG-01 cells, treatment with BIBR1532, Dox, or Bor as single agents led to obvious suppression of cell viability and facilitation of apoptosis (p < 0.01). When administered in combination, cell viability was further inhibited, while apoptosis was further facilitated, compared with Dox or Bor alone (p < 0.05). Consistently, the treatment time was also important for cell viability, as cells cultured for 48 h with the compounds exhibited lower viability than those cultured for 24 h. In addition, western blotting showed that the expression of Bcl-xL and Survivin proteins decreased moderately, whereas the expression of Bad protein increased significantly after K562 and MEG-01 cells were stimulated with BIBR1532, Dox, or Bor alone (p < 0.05, Fig. 6C). Relative to Dox or Bor alone, co-treatment with BIBR1532 further reduced Bcl-xL and Survivin protein expression while increasing Bad protein expression.

Figure 6 (A–C) Combined effect of BIBR1532 with Dox or Bor on suppressing growth but inducing death in multiple myeloma cells.

Combination of Dox or Bor with BIBR1532 synergistically inhibited TERT and c-MYC expressions in MM cells

To verify the inhibitory effect of the combination on telomerase activity, TERT and c-Myc expression was assessed at both the mRNA and protein levels. Compared with Dox or Bor alone, combined treatment with BIBR1532 resulted in a significant decrease in TERT expression in both K562 and MEG-01 cells (p < 0.05, Figs. 7A–7B). Furthermore, compared with each single-compound treatment, the combination of BIBR1352 produced a marked decrease in c-MYC protein expression in K562 and MEG-01 cells (p < 0.05).

Figure 7 (A–B) The synergistic effect of BIBR1532 with Dox or Bor on the inhibition of TERT and c-MYC expression in multiple myeloma cells.

Combination of Dox or Bor with BIBR1532 synergistically suppressed the PI3K/AKT/mTOR pathway while facilitating the ERK1/2 MAPK pathway in MM cells

Western blotting was conducted to explore the function of the combination on the PI3K/AKT/mTOR and ERK1/2 MAPK pathways. As shown in Fig. 8, for both K562 and MEG-01 cells, compared with Dox or Bor alone, co-administration with BIRI1532 caused a significant decrease in PI3K, AKT, and mTOR phosphorylation leading to an evident upregulation in ERK1/2 phosphorylation (p < 0.05). In addition, compared with Dox or Bor alone, the combination with BIBR1532 upregulated MAPK phosphorylation in MEG-01 cells (p < 0.05).

Figure 8 (A–B) BIBR1532 combined with Dox or Bor worked synergistically in suppressing the PI3K/AKT/mTOR pathway but facilitating the ERK1/2 MAPK pathway in multiple myeloma cells.

Discussion

Uncontrolled cell proliferation and apoptosis contribute to the initiation and development of most cancers. Thus, medical researchers believe that suppressing cell proliferation and facilitating apoptosis can act as starting points for cancer treatment (Mao et al., 2021). In the present study, BIBR1532 inhibited telomerase activity to repress cell viability and promote apoptosis in MM cells by suppressing the PI3K/AKT/mTOR and ERK1/2 MAPK pathways. Furthermore, BIBR1532 could exert a synergistic effects with Dox and Bor against MM cells. Thus, BIBR1532 is a potential therapeutic agent for MM.

Telomerase is broadly expressed in cancer cells but is almost completely inactivated in healthy somatic cells (Zhao et al., 2018). Some experiments have revealed that telomerase sustains the proliferative ability of tumor cells and promotes tumor development by regulating telomere length (Prado et al., 2021). If telomerase is inactivated, cells progress to a proliferative quiescent state and die (Volleth et al., 2020). Telomerase is vital for cancer progression and has been proposed as a promising biomarker for diagnosing and treating cancer (Farid Aql et al., 2021). A previous study reported that telomerase activity and expression are linked to poor survival in MM (Shalem-Cohavi et al., 2019). Hence, the inhibition of telomerase activity has great therapeutic potential against MM. Numerous telomerase inhibitors have been developed for this purpose. BIBR1532, a synthetic telomerase inhibitor, reduces the viability of various tumor cells (Yang et al., 2017). As shown in previous studies, BIBR1532 sensitizes breast cancer cells to natural killer (NK) cell therapy (Mazloumi et al., 2023). Ding et al. (2019) reported that BIBR1532 efficiently enhanced radiosensitivity in non-small cell lung carcinoma (NSCLC) cells. Consistent with previous research, this study demonstrated that BIBR1532 suppressed cell growth and induced cell death in K562 and MEG-01 cells by reducing the expression of anti-apoptotic proteins (such as Bcl-xL and Survivin), and increasing pro-apoptotic protein expression (Bad).

hTERT is an important component that determines telomerase activity and is upregulated in various cancers including MM (Ameri et al., 2019). Several lines of evidence suggest that multiple transcription factors in the hTERT promoter participate in the regulation of hTERT expression (Lai et al., 2007). Among these transcription factors, c-MYC is a powerful regulator for hRERT transcription and expression (Collins, 2008). Of note, activated c-MYC leads to MM pathophysiology, and c-MYC activity is upregulated during MM development (Holien et al., 2015). Thus, the inhibitory effect on telomerase activity was validated by measuring TERT and c-MYC expression. In this study, we found that there was decreased TERT and c-MYC expression upon BIRI1532 treatment. There is evidence that hTERT sustains telomere length, which determines cellular proliferative capability (Prado et al., 2021), and that silencing hTERT expression can directly induce cell apoptosis (Barczak et al., 2018). Therefore, we speculated that BIBR1532 may exhibit anti-proliferative and pro-apoptotic effects by inhibiting telomerase activity.

The PI3K/AKT/mTOR and ERK1/2 MAPK pathways are crucial for modulating the malignant phenotypes of tumor cells, including cell survival (Li et al., 2019) and apoptosis (Lee et al., 2018). In the PI3K/AKT/mTOR pathway, AKT activation facilitates tumor invasion and metastasis by phosphorylating mTOR (Gao et al., 2020). Furthermore, cell proliferation-associated protein (P70S6K) and cell cycle-related protein (Cyclin D1) are direct downstream targets of the PI3K/AKT/mTOR pathway (Zhao & Hu, 2019). Activation of the MAPK pathway is believed to promote cell apoptosis by phosphorylating multiple pro-apoptotic effectors, whereas the ERK1/2 pathway is associated with cell proliferation (Li et al., 2018). The PI3K/AKT/mTOR pathway is activated, whereas the ERK1/2 MAPK pathway is inactivated in different types of tumor tissues (Hsieh et al., 2014). Tahtouh et al. found that BIBR1532 reduced the alpha-fetoprotein (AFP) concentration (a diagnostic marker for hepatocellular carcinoma) by inhibiting the PI3K/AKT/mTOR pathway (Tahtouh et al., 2015). Considering the fundamental effects of the PI3K/AKT/mTOR and ERK1/2 MAPK pathways in MM pathogenesis, western blotting was conducted to examine whether BIBR1532 affects the expression of the pathway-related proteins. In line with previous findings, this study also demonstrated that, upon treatment with BIBR1532, the PI3K/AKT/mTOR pathway was repressed, whereas the ERK1/2 MAPK pathway was activated in the MM cells. These results imply that BIBR1532 may exert an antitumor effect on MM by suppressing cell proliferation and promoting apoptosis via the PI3K/AKT/mTOR and ERK1/2 MAPK pathways.

MM development involves many complex mechanisms as well as pathways. Alone, one drug is weak in terms of treating the disease and prone to inducing drug resistance; hence, combination drugs have been proposed as a promising modality for MM treatment (Jelonek et al., 2021). In addition to overcoming drug resistance, another goal of using compounds in combination is to achieve synergistic therapeutic efficacy (Lang et al., 2019). In this study, BIBR1532 was combined with two common drugs (Dox and Bor) to determine whether the combination could achieve a synergistic therapeutic effect against MM in vitro. The results showed that this combination was more effective than Dox or Bor alone at suppressing cell viability, inducing apoptosis, decreasing telomerase activity, inhibiting the PI3K/AKT/mTOR pathway, and facilitating the ERK1/2 MAPK pathway.

Despite the significant contributions of this study, it has some limitations. First, normal cell lines were absent from the study, and U266 cells are more commonly employed for MM research. In addition, we did not use agonists or inhibitors to further validate the critical role of the PI3K/AKT/mTOR and ERK1/2 MAPK pathways in the reduction of telomerase activity by BIBR1532. In the future, normal cell lines (such as HS-5, human bone marrow stromal cells), U266 cells, PI3K/AKT/mTOR pathway agonists, and ERK1/2 MAPK pathway inhibitors will be used to further verify the findings of this study.

Conclusions

In summary, this study found that BIBR1532 can repress proliferation and facilitate apoptosis in MM cells, and that the specific mechanism is related to the PI3K/AKT/mTOR and ERK1/2 MAPK pathways. Moreover, BIBR1532 combined with Dox and Bor may act synergistically to treat MM by modulating the PI3K/AKT/mTOR and ERK1/2 MAPK pathways. These findings will help us to better understand the effects and mechanisms of BIBR1532 in MM treatment. BIBR1532 may be a crucial drug in the treatment of MM.

Supplemental Information

Data S1 Raw data

Click here for additional data file.

Supplemental Information 2 The original gels of western blot

Click here for additional data file.

Additional Information and Declarations

Competing Interests

Author Contributions

Data Availability

The authors declare there are no competing interests.

Yuefeng Zhang conceived and designed the experiments, authored or reviewed drafts of the article, and approved the final draft.

Xinxin Yang performed the experiments, authored or reviewed drafts of the article, and approved the final draft.

Hangqun Zhou analyzed the data, authored or reviewed drafts of the article, and approved the final draft.

Guoli Yao analyzed the data, prepared figures and/or tables, and approved the final draft.

Li Zhou analyzed the data, prepared figures and/or tables, and approved the final draft.

Chunyan Qian analyzed the data, authored or reviewed drafts of the article, and approved the final draft.

The following information was supplied regarding data availability:

The raw data and the original gels of western blot are available in the Supplemental Files.

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
