# Peer review of "BIBR1532 inhibits proliferation and enhances apoptosis in multiple myeloma cells by reducing telomerase activity"

_PeerJ, doi:10.7717/peerj.16404_

## Round 0.1 · original submission · Major Revisions

Please carefully read the comments from the reviewers and address all questions. The language also needs to be carefully edited.

**Language Note:** The Academic Editor has identified that the English language must be improved. PeerJ can provide language editing services - please contact us at [email protected] for pricing (be sure to provide your manuscript number and title). Alternatively, you should make your own arrangements to improve the language quality and provide details in your response letter. – PeerJ Staff

Reviewer 1 ·

Basic reporting

Zhang et al. showed that BIBR1532 can inhibit multiple myeloma cells through regulating telomerase activity. Overall, the findings are interesting. However, there are some concerns that need to be addressed, which are listed as follows:

1. Some grammar errors have been identified. The authors should seek help from certified English Editing Service to polish its English. For example, the sentence “K562 and MEG-01 cells were cultured with BIBR1532 for different concentrations, after 24 and 48 h, cell survival was examined” and other sentences. Maybe you can break down the long sentences to short ones.

In the abstract, sometimes the authors used “Dox” “Bor”, sometimes they used “doxorubicin” or “bortezomib”

In Line56, change “have reported” to “reported”, which is suitable for other similar sentences.

2. What is the full name of “MMPs” in Line 65?

3. It is better that the authors can provide the primers for RT-PCR of telomerase.

4. The authors should provide the molecular weight for all WB blots.

5. In Figure5, the authors showed that BIBR1532 upregulated ERK1/2 phosphorylation, which is opposite with previous study as the authors stated in Line 65. How to explain this?

6. In Line213, Fig.8A-8B should be Fig.7A-7B.

Experimental design

The experimental design is ok.

Validity of the findings

Overall, the findings are interesting

Reviewer 2 ·

Basic reporting

In this study, authors explore the effect of BIBR1532 in multiple myeloma (MM) in vitro. The results showed that BIBR1532 inhibited K562 and MEG-01 cellular survival in a dosage- and time-dependent fashion. In addition, BIBR1532 hindered cell proliferation while promoting apoptosis, and the effect was enhanced with increased BIBR1532 concentration. Moreover, BIBR1532 inhibited TERT and c-MYC expressions, AKT and mTOR phosphorylation, and facilitated ERK1/2 and MAPK phosphorylation. Additionally, BIBR1532 combined with doxorubicin or bortezomib showed synergistic effects in MM treatment.

Experimental design

Although the work provides a useful perspective on MM treatment, there are several suggestions to improve the article:
Although the work provides a useful perspective on MM treatment, there are several suggestions to improve the article:
1. Please provide more details about BIBR1532 in the Introduction section.
2. The study did not use normal cells (such as HS-5 cells) as a control group to determine whether BIBR1532 is toxic to normal cells, which should to be supplemented or mentioned or mentioned as a limitation in the discussion.

Validity of the findings

1. Since the authors believed that BIBR1532 can inhibit the PI3K/AKT/mTOR pathway, it would be appropriate to employ Western blot to detect the expression levels of PI3K and p-PI3K in Figure 5 and Figure 8.
2. The protein expression was not statistically analyzed in Figure 6C.
3. Figure 7 is missing in the results section, please check.

Additional comments

1. Abbreviations should be written in full when they appear first.
2. There are some grammatical errors in the article, which need to be carefully corrected throughout the manuscript.
3. As the authors indicate the critical role of the PI3K/AKT/mTOR pathway and ERK1/2 MAPK pathway in reducing telomerase activity by BIBR1532, further study focused on these pathways is need to clarify the anti- multiple myeloma mechanism of BIBR1532.

---

## Round 0.2 · Minor Revisions

I did not find the point-by-point responses to the comments and suggestions from the reviewers. Please also attach the certificate for English language editing.

**Language Note:** The Academic Editor has identified that the English language must be improved. PeerJ can provide language editing services - please contact us at [email protected] for pricing (be sure to provide your manuscript number and title). Alternatively, you should make your own arrangements to improve the language quality and provide details in your response letter. – PeerJ Staff

Reviewer 1 ·

Basic reporting

The authors have well addressed my concerns.

Experimental design

The authors have well addressed my concerns.

Validity of the findings

The authors have well addressed my concerns.

Reviewer 2 ·

Basic reporting

No comment.

Experimental design

No comment.

Validity of the findings

No comment.

Additional comments

No comment.

---

## Round 0.3 · accepted · Accept

The authors have addressed the reviewers' concerns and provided their manuscript with track changes. They also provided the certificate for English language editing in the attachment. The paper may be acceptable at the current stage.